# Effects of Feeding Different Levels of Sprouted Barley on Fermentation Characteristics, Bacterial Quantification, and Rumen Morphology of Growing Lambs

**DOI:** 10.3390/vetsci10010015

**Published:** 2022-12-26

**Authors:** Abdulrahman S. Alharthi, Hani H. Al-Baadani, Mutassim M. Abdelrahman, Ibrahim A. Alhidary

**Affiliations:** Department of Animal Production, College of Food and Agriculture Science, King Saud University, P.O. Box 2460, Riyadh 11451, Saudi Arabia

**Keywords:** bacterial, digestibility, growing lamb, rumen morphology, sprouted barley, volatile fatty acid

## Abstract

**Simple Summary:**

Sprouted barley has been proposed as one of the solutions to the feed challenge faced by livestock producers. The current hypothesis is that replacing traditional feed for growing lambs with sprouted barley could develop the rumen ecosystem by improving nutrient digestibility, fermentation through change in volatile fatty acid, and some bacteria. Therefore, this study aims to determine the effect of feeding sprouted barely on the performance, digestibility, rumen fermentation profile, bacterial quantification, and rumen morphology of lambs. Sprouted barley improved digestibility and rumen histomorphometric and increased the concentration of some volatile fatty acids and rumen bacteria, but decreased dry and organic matter intake, negatively affecting the weight gain of lambs fed 100% with sprouted barley.

**Abstract:**

The objective of the present study was to investigate the effects of sprouted barley inclusion level on the growth performance, digestibility, volatile fatty acids, bacterial quantification, and rumen morphology of growing lambs. Five dietary treatments with sprouted barley (0, 25, 50, 75 and 100%) and nine replicates per dietary treatment were performed on forty-five Awassi lambs (90 days old). The average weight gain, intake, and digestibility of dry and organic matter were recorded. The pH, color, volatile fatty acids, bacterial quantification, and rumen histomorphometry were also determined. The results showed that the average dry and organic matter intake in T2 to T4 and the average weight gain in T4 decreased linearly. In contrast, the digestibility of dry and organic matter by sprouted barley (T2 to T4) was higher. The pH values and rumen color were not affected. Concentrations of formic acid, acetic acid, butyric acid, and the ratio of acetic acid to propionic acid were increased, while lactic acid and total volatile fatty acids were lower in all levels of sprouted barley. In addition, lambs fed T4 had a higher quantification of *Anaerovibrio Lipolytica*, *Butyrivibrio Fibrisolvens*, and *Streptococcus Bovis* quantification. *Selenomonas Ruminantium* was higher in T1, T2, and T4, whereas *Megashpaera Elsdenii* was lower in T1 to T3. The rumen histomorphometric was improved by sprouted barley (T2 and T3). Sprouted barley improved digestibility and rumen histomorphometry and increased the concentration of some volatile fatty acids and rumen bacteria but resulted in a decrease in average dry and organic matter intake, which negatively affected weight gain in lambs fed 100 % sprouted barley. Further studies are required to determine the potential effects on growing lambs fed sprouted barley.

## 1. Introduction

Feed is considered the most important and expensive component of raising sheep, so there is usually a need to evaluate sustainable feed sources, including forages and their by-products [1]. Recently, hydroponics, including sprouted barley, has been proposed as one of the solutions to produce animal feed in a short time, especially in areas suffering from water scarcity, low rainfall, and seasonal variations in feed with the inflation in feed prices, which pose significant challenges to livestock producers [2,3]. In addition, the rising cost of feed stimulated many researchers and animal production producers to look for an alternative solution for feed production, such as using sprouted barley technology to meet nutritional needs and improve animal performance [4]. Therefore, sprouted barley is a new way of forage production, characterized by a short growing season, year-round production, the ability to manage the environment of the plants, and high nutrient content [5].

Rumen bacteria are critical to the fermentation process, which mainly contributes to the digestion and conversion of forages into volatile fatty acids (VFAs) and microbial proteins [6,7]. While rumen bacteria are stable, the type of diet given to animals may result in variations in the rumen ecosystem [8,9]. Sprouted barley is a potential source of nutrients for the activity of rumen bacteria that improve their digestibility [10]. Consequently, hydroponics of barley resulted in improved nutrient content, including crude protein, fiber, fat, sugars, minerals, and vitamins, and decreased dry matter, starch, and antinutrients [11,12]. These biochemical changes in barley grains during hydroponics may improve digestibility due to the activities of hydrolytic enzymes [13]. Several studies reported that lambs fed sprouted barley had a higher dry matter and organic matter digestibility [11,14]. Replacement of sprouted barley increased total VFA and propionic acid concentrations in lambs [15,16]. Farghaly et al. [17] indicated that germinated barley promoted carbohydrate fermentation in the rumen. 

Studies investigating the replacement of traditional feed with sprouted barley (barley grain and alfalfa hay) on the rumen ecosystem are limited and need further investigation to determine the optimal levels in lambs. The hypothesis of the present study is that replacement of sprouted barley compared to traditional feed could be a possible strategy for rumen ecosystem development by improving nutrient digestibility, fermentation through the change in volatile fatty acid, some bacterial activity, and rumen morphology. This study aimed to determine the overall effect of feeding sprouted barely on the performance, digestibility, rumen fermentation profile, bacterial quantification, and rumen morphology of growing lambs.

## 2. Materials and Methods

### 2.1. Animal Welfare and Ethics Clearance

The study was conducted at King Saud University, Riyadh, Saudi Arabia. The use of animals and the procedures adopted in this study followed the Animal Welfare Act of Practice for the Care and Use of Animals for Scientific Purposes and were approved by the Research Ethics Committee, King Saud University (REC-KSU; Ethics Reference No: KSU-SE-22-01).

### 2.2. Diets and Management Practices

Natural alfalfa hay was purchased and cut to a length of approximately 6–8 cm to minimize preferential selection. It was mixed with barley grains (30% alfalfa hay:70% barley grain) and offered ad libitum to individual lambs as a basal diet (traditional diet). Hydroponic cultivation of locally obtained barley seeds was carried out in the Department of Plant Production, College of Food and Agricultural Sciences, King Saud University, Saudi Arabia, according to the method described by Al-Baadani et al. [3]. The hydroponic chamber was designed to hold 140 trays (70 × 30 cm) with a capacity for seven growth stages (7 days). Therefore, the production cycle of sprouted barley was continued daily during the study period to feed lambs at different levels as an alternative to traditional feed, after the cut, for easy consumption, without picking only the leaves. 

### 2.3. Animals and Management Practices

This study was conducted for 75 days at the farm of the Department of Animal Production, College of Food and Agricultural Sciences, King Saud University, Saudi Arabia. Before starting the actual study, all lambs were acclimatized for 14 days to the used diets and vaccinated against common diseases (manufactured by Ibrize Co.-KSA) according to the protocol recommended by the Animal Resources Directorate, Ministry of Environment, Water and Agriculture [18]. Forty-five growing Awassi male lambs, aged approximately 3 months (27.85 ± 2.5 kg), were randomly assigned to five dietary treatments individually (nine lambs per dietary treatment): CON = traditional diet 100% (barley 70%: alfalfa hay 30%), T1 = traditional diet 75% plus sprouted barley 25%, T2 = traditional diet 50% plus sprouted barley 50%, T3 = traditional diet 25% plus sprouted barley 75%, and T4 = sprouted barley 100%. Nutrient composition on a dry matter basis of dietary treatments is shown in Table 1. All lambs were fed twice daily (08:00 and 14:00) and had ad libitum access to feed and water throughout the study period.

### 2.4. Feed Analyses

Feed samples from each treatment were collected before the study and then once per month until the end of the study, mixed uniformly, dried (60 °C) to determine initial moisture content, and ground to a fine powder and stored at −20 °C until the nutrient composition was approximately analyzed at King Saud University laboratories. Dry matter content was determined by drying samples in an oven at 100 °C for 24 h; while ash content was determined by incinerating samples at 550 °C for 3 h in a muffle furnace. Crude protein was measured following the AOAC method [19]. Neutral detergent fiber and acid detergent fiber were determined according to methods described by Van Soest et al. [20]. 

### 2.5. Growth Performance and Digestion Trial

All lambs were weighed on day 1 and day 75 during the study period to determine the average weight gain. Live weight gain (LWG) was calculated as the difference between final and initial weight [3]. Dry matter and organic matter intake (DMI and OMI) were estimated as the difference between rejected and offered feed divided by 75 days and then multiplied by the percentage of dry matter or organic matter [11]. Apparent total tract digestibility of dry matter (DM) and organic matter (OM) were estimated as the difference between intake and fecal excretion divided by intake and multiplied by one hundred, according to Mekonnen et al. [21]. 

### 2.6. Rumen Fermentation Characteristics

On days 45 and 75, rumen fluid samples were collected from all forty-five animals (nine lambs per dietary treatment) before and after morning feeding (0 and 3 h) by gently inserting a stomach tube into the rumen through the mouth. The rumen fluid was aspirated using a vacuum pump (KNF Neuberger, Freiburg, Germany) according to the method of Farghaly et al. [17]. Rumen fluid pH was determined immediately in duplicate using a digital pH meter (model pH 211; Hanna Instruments, Woon-socket, RI, USA). Rumen fluid samples were collected after morning feeding (3 h) at 75 days and filtered through a cloth into a sterile 30 mL plastic tube, and 0.3 mL of H_2_SO_4_ was added to stop fermentation. All samples were stored at −20 °C until volatile fatty acids were analyzed by the method described by Szulc et al. [22]. A mixture of volatile fatty acids (VFAs) containing lactic, formic (C1), acetic (C2), propionic (C3), and butyric (C4) acids as an internal standard were purchased from Dr. Ehrenstorfer (Augsburg, Germany). VFA concentrations were determined by high-performance liquid chromatography (Agilent Technologies, 1260 series, Palo Alto, CA, USA). Each volatile fatty acid concentration’s results were reported as a percentage of total VFAs.

### 2.7. Rumen Tissue Color Measurements

Rumen tissue color measurements were determined in duplicate immediately after the slaughter at the end of the study (75 days) using a Minolta Chroma Meter (Konica Minolta, CR-400-Japan) with a CIELAB color system for color values (L* = value denotes brightness; a* = redness and b* = yellowing) as previously described by Alhidary et al. [23].

### 2.8. Ruminal Bacterial Quantification

At slaughter (75 days), approximately 0.5 mL of rumen fluid was collected from each lamb, placed in sterile 1.5-mL Eppendorf tubes, and snap-frozen in liquid nitrogen and stored at −80 °C until DNA extraction. All samples were centrifuged at 10,000× *g* for 10 min at 4 °C to remove the supernatant containing rescued solids. Total DNA was extracted step by step using QIAamp DNA Stool Mini kit (Qiagen, Germantown, MD, USA) according to the manufacturer’s recommendations. The quantity and quality of DNA were determined using a Nanodrop spectrophotometer (Thermo Scientific, 2000 Nanodrop, Waltham, MA, USA). Quantitative PCR (qPCR) of DNA samples was performed on an Applied Biosystems Step One Real-time PCR System (7300 Real-Time PCR System, Applied Biosystems) using Power SYBR^®^ Green PCR Master Mix (Applied Biosystems, Thermo Fisher Scientific, Foster, CA, USA) with target primers from eight bacteria (Table 2). Each reaction was performed in duplicate for each target gene. Quantification of microbiota in each sample was calculated by comparison with a standard curve generated for serially diluted pool DNA at a known concentration (from 10^2^ to 10^12^ copies/mL of rumen fluid) according to Li et al. [24] and Izuddin et al. [25]. The microbiota quantification result was expressed as Log10 microbial per mL of rumen fluid. 

### 2.9. Rumen Histomorphometric

After slaughter, tissues were collected from the mid-portions of the rumen (1 cm^2^) from nine lambs per dietary treatment and then fixed in neutral buffered formalin (10%). Histological sections were automatically prepared using a tissue processor (Tissue-Tek VIP 5 Jr, Sakura, Japan), including dehydrating by gradations in ethyl alcohol and embedding in paraffin wax. Sections were then cut onto slides using a microtome system (Leica Biosystems, RM 2255, Wetzlar, Germany) and stained with eosin and hematoxylin (Leica, CV5030, Wetzlar, Germany) according to the method described by Alhidary et al. [23]. All histomorphometric, such as papilla height, papilla width, stratum corneum (SC), the width of the epithelium (WE), lamina propria (LP), and submucosa (SM), were measured in five photomicrographs per animal (45 per treatment) using a light microscope (Nikon, Corp., Japan) together with camera software for image analysis (AmScope digital camera-attached Ceti England microscope, Irvine, CA, USA) according to Alharthi et al. [26]. Papilla density (number/cm_2_) was estimated using a video camera. The surface area (SA = 2π × (papilla width/2) × papilla height) and total papilla surface area (TSP = papilla height × papilla width × 2 × density) were calculated according to Abdelrahman et al. [27].

### 2.10. Statistical Analysis

All lambs were randomly divided into experimental units for dietary treatments using a completely randomized design (CRD). All data obtained from this study were analyzed based on a one-way analysis of variance (ANOVA) using the general linear models (GLM) in SAS 9.4 software [28]. The following statistical model was used: Observed values (Yij) = general mean (μ) + dietary treatments as main effect (Ti; I = CON, T1, T2, T3, and T4) + the random error (eij). Rumen fluid pH was analyzed using the SAS procedure [29] with repeated measurements to determine sampling time and interaction between dietary treatments and time points (sampling at 45 and 75 days or before and after feeding at 0 and 3 h per time point). A Duncan multiple range test was used to compare the means of the dietary treatments when the effect of the dietary treatment was considered statistically significant at *p* < 0.05. In addition, linear or quadratic responses were analyzed by orthogonal contrasts (CON vs. T1, T2, T3, and T4). All means were presented with the standard error of the mean (means ± SEM). A Pearson correlation between rumen bacteria metabolites and rumen pH, papilla height, and total papilla surface area was calculated according to Belote et al. [29].

## 3. Results

The effects of dietary treatments on growth performance and apparent total tract digestibility of the growing lambs are shown in Table 3. The results of the study show that the partial replacement of sprouted barley (T1 to T3) had no negative effect on LWG, while lambs fed 100% sprouted barley (T4) had a lower LWG compared to the traditional diet (CON; *p* < 0.05). DMI and OMI were lower at T2 and T3, followed by T4, compared to T1 and CON (*p* < 0.05). Apparent total tract digestibility of DM and OM was increased in dietary treatments from T2 to T4 compared to CON, while there was no significant difference between T1 and CON (*p* < 0.05). Moreover, a linear response of LWG, DMI, OMI, and apparent total tract digestibility (DM and OM) was observed with the replacement of sprouted barley (*p* < 0.05).

The effects of dietary treatments on pH and color measurements in the rumen of lambs are shown in Table 4. On days 45 and 75, rumen pH before and after feeding (0 and 3 h) was not affected by dietary treatments or time and the interaction (*p* > 0.05) and showed no linear or quadratic response to the replacement of sprouted barley (*p* > 0.05). Measures of rumen color such as lightness (L*) and redness (a*) were not affected by dietary treatments (*p* > 0.05), whereas yellowing (b) was lower at T4 compared to CON and the other dietary treatments (T1 to T3; *p* < 0.05). In addition, there was no linear or quadratic response to a* and b* (*p* > 0.05), but L* showed a quadratic response with the replacement of sprouted barley (*p* < 0.05).

The effects of dietary treatments on the concentrations of volatile fatty acids in the rumen of lambs are shown in Table 5. On the 75th day of the study period, after feeding (3 h), the results show that the relative lactic acid concentration was lower in T3, followed by T1, T2 and T4, compared to CON (*p* < 0.05). Formic acid was higher in T4 than in T1, T2 and T3 compared to CON (*p* < 0.05). Dietary treatments T2 and T4, followed by T1 and T3, increased acetic acid concentration more than CON (*p* < 0.05). The sprouted barley treatments (T2 and T3) decreased propionic acid concentration compared to CON and the other diet treatments (*p* < 0.05). On the other hand, butyric acid concentration was higher in T3 than in T1 and T2, while T4 showed no effect compared to CON (*p* < 0.05). The acetate-to-propionate ratio (C2/C3) was higher in T2, followed by T3 and T4 compared to CON (*p* < 0.05). Lambs fed 100% sprouted barley (T4) had lower total VFA concentrations, followed by T2, then T1 and T3 compared to CON (*p* < 0.05). In addition, a linear response of lactic acid and TVFA and a quadratic response of formic, acetic, propionic, butyric acids and C2/C3 ratio were observed when sprouted barley was replaced (*p* < 0.05).

The effects of dietary treatments on the quantification of rumen bacteria are shown in Figure 1. When lambs received 100% sprouted barley (T4), quantification of *Anaerovibrio lipolytica* and *Streptococcus bovis* were increased compared to the other treatments and CON (*p* < 0.05). Dietary treatments T1, T2, and T4 increased the quantification of *Butyrivibrio Fibrisolvens* and *Selenomonas Ruminantium* compared with CON (*p* < 0.05). At the same time, *Megashpaera Elsdenii* was reduced in lambs receiving T1 to T3 compared with T4 and CON (*p* < 0.05). *Fibrobacter succinogenes*, *Ruminocoocus Albus*, *Ruminococcus Flavefaciens,* and total bacteria showed no effect in all dietary treatments compared with CON (*p* > 0.05). In addition, there was a linear response to sprouted barley replacement in *Anaerovibrio Lipolytica* and *Selenomonas Ruminantium* and a quadratic response in *Megashpaera Elsdenii* (*p* < 0.05), but other quantifiable bacteria showed no linear or quadratic response to sprouted barley replacement (*p* > 0.05).

The effects of dietary treatments on lamb rumen histomorphometry are shown in Table 6. The papilla height, papilla width, and SA of papillae increased in lambs that received dietary treatments (T2 and T3) and decreased when lambs received 100% sprouted barley (T4) compared with CON (*p* < 0.05). Replacing the traditional diet with 75% sprouted barley (T3) had the lowest papilla density compared with T1 and T4 but was not affected compared with T2 and CON (*p* < 0.05). Therefore, TSP was greater in sprouted barley (T1 to T3) than in T4 and CON (*p* < 0.05). Lambs fed 100% sprouted barley (T4) had a higher SC than the other dietary treatments (*p* < 0.05). The WE was higher at T2, followed by T1, T3, and T4 compared to CON (*p* < 0.05). Dietary treatment of T3 increased the LP and SM more than CON (*p* < 0.05). In addition, all histomorphometric measurements showed a quadratic response (*p* < 0.05), except for papilla density and stratum corneum, which were neither linearly nor quadratically affected by the dietary treatments (*p* > 0.05). An example of a representative rumen histomorphometric from each dietary treatment is shown in Figure 2.

The results of correlations between volatile fatty acids and pH values or rumen histometeric are shown in Table 7. A strong positive correlation was observed between lactate and TVFA with L, while a negative correlation between format and propionate with L and between format and TSP (*p* < 0.05). However, rumen bacteria metabolites showed no correlations (*p* > 0.05) with rumen pH values.

## 4. Discussion

The results of the current study showed that the replacement of sprouted barley (T1 to T3) had no negative effect on LWG compared with a traditional diet (CON), whereas lambs fed 100% sprouted barley (T4) had the lowest LWG. This decrease in LWG was due to a negative linear response in DMI and OMI with the increase in the replacement of sprouted barley instead of the traditional diet, which could be due to the lower DM content of sprouted barley (Table 1). Sprouted barley has a lower DM content, which means that lambs may not be able to consume more dry matter-based feed, which may affect growth performance [30]. These results are in agreement with those of Muhammad et al. [31], who reported that the replacement of sprouted barley might affect growth performance compared to a traditional diet. This could be due to consuming less DM without meeting the nutrient requirements for peak performance. However, our results show that the apparent total tract digestibility of DM and OM increased linearly when the proportion of sprouted barley was increased. This could be due to rumen’s easy degradation of sprouted barley in the rumen. These results are consistent with the fact that lambs fed sprouted barley had higher DM and OM digestibility than lambs fed barley seed [11,14]. Ikram et al. [13] showed that several biochemical changes in barley seeds during hydroponics can improve digestibility due to the activities of hydrolytic enzymes that improve the nutrient content of barley seeds, including protein, fiber, fat, sugar, and minerals, while reducing antinutrients. 

Rumen pH before and after feeding was not affected by the replacement of sprouted barley compared to a traditional diet. Rumen pH values in the current study (6.2 to 7.0) were within the acceptable range of microbial digestive activities. A healthy rumen should have a pH between 6.0 and 7.0. Therefore, Al-Saadi and Al- Zubiadi [14] reported that any change in rumen pH might indicate a change in the rumen ecosystem, such as a high level of microbial activity or an excessive concentration of VFAs. Tawfeeq et al. [5] in lambs and Helal [32] in ewes showed that substitution of sprouted barley (25% to 100%) with a traditional diet (green alfalfa) had no effect on rumen pH at the end of the study, while in the middle of the study rumen pH decreased. This could be due to the fact that the sprouted barley root layer contains barley grains that are 55%–75% starch, resulting in an increase in fermented carbohydrates to volatile fatty acids and lowering rumen pH. However, since there is no difference in rumen pH when sprouted barley is substituted compared to the traditional diet, this could be the result of an adjustment in the rumen environment and increased saliva production by the sprouted barley. The current study showed that rumen color measurements were not affected except for yellow coloration (b*), which was lower in lambs fed 100% sprouted barley (T4) compared to the traditional diet (CON).

Our results show that lambs fed sprouted barley had lower lactic acid (T1 to T4), propionic acid (T2 and T3), and TVFA (T1 to T4) concentrations than lambs fed CON. This could be due to the high digestibility of sprouted barley, which can pass through the rumen more quickly, and thus the rapid utilization of VFA for maintenance and production. These results are similar to those of Hafla et al. [4], who demonstrated a lower concentration of VFAs in the rumen of lambs fed sprouted barley compared to CON. However, our results contradict those of Dung et al. [15,33], who reported that VFA concentrations were increased when sprouted barley was substituted in lambs. In contrast, the concentrations of formic acid and acetic acid were higher in sprouted barley (T1 to T4) than in CON. T3 had a higher concentration of butyric acid, while T4 showed no effect compared to CON. Mpanza et al. [16] showed that the replacement of sprouted barley increased propionic acid concentration in growing lambs. Since propionic acid is ruminant’s main source of energy [34], this suggests that T1 and T4 increased energy availability to the animals more than the T2 and T3 diets. The C2/C3 ratio was higher for T2 to T4 than for CON and lower for T1, T3, and T4 than for T2. Ma et al. [35] indicated that a lower C2/C3 ratio provides more energy to the host animal, which affects growth. High acetic acid concentrations in T2 and T4, followed by T1 and T3, compared to CON might indicate that germinated barley promotes carbohydrate fermentation in the rumen. This is consistent with the results of Farghaly et al. [17] who showed that sheep fed sprouted barley had a higher concentration of acetic acid in the rumen than the control group.

The animal’s rumen is a dynamic, diverse, and complex compartment due to bacterial consortia. Rumen bacteria are crucial for the fermentation process, which mainly contributes to the digestion and conversion of forages into VFAs and microbial proteins [6,7]. The main source of energy for the animal is the VFAs formed in the rumen [36]. While rumen bacteria are stable, changes in diet may be associated with variations in the number of rumen bacteria [8]. Therefore, feed utilization depends on the development of the rumen ecosystem [9,37]. Barley germination increased hydrolytic enzyme activity, which improved the content of crude protein, fat, essential amino acids, and sugars [11]. Thus, rumen bacteria can ferment these non-fibrous carbohydrates in addition to starch and sugar. In the present study, increased activity of microbial species such as *Anaerovibrio lipolytica* (lipolytic bacterium), *Streptococcus bovis* (amylolytic bacterium), *Butyrivibrio fibrisolvens* (cellulolytic bacterium), and *Selenomonas ruminantium* (saccharolytic bacterium) was measured in lambs fed sprouted barley compared to CON. These results agree with those of Hafla et al. [4], who reported that ruminants fed sprouted barley had an increased rumen bacterial profile. In contrast, the rumen bacterial profile of lambs decreased when sprouted barley was added at different levels [38]. Adding sprouted barley (T1 to T3) to the traditional diet decreased the quantification of *Megasphaera Elsdenii* (acid metabolizers) compared to T4 and CON. These results may indicate that sprouted barley increased bacterial activity in the rumen and increased the digestibility of sprouted barley content. Most cellulolytic bacteria, including *Fibrobacter succinogenes*, *Ruminocoocus albus*, and *Ruminocoocus flavefaciens*, had no effect in any dietary treatments compared to CON.

Histomorphometric rumen parameters such as papilla height, width, SA, density, TSP, SC, WE, LP, and SM are most commonly used to evaluate rumen development and health in growing lambs [39]. The current study showed that papilla height, width, and SA increased when lambs were fed sprouted barley (T2 and T3) and decreased in T4 compared to CON. Most studies report rumen papilla height and width as estimates of rumen epithelial growth [40]. The increased SA reflects the increased availability of the surface for the absorption of VFAs [41]. Consequently, our results showed that TSP was greater in sprouted barley (T1 to T3) than T4 and CON. Shen et al. [42] reported that diets with high digestibility increase in rumen papillae in goats. Additionally, lambs fed 100% sprouted barley (T4) had higher stratum corneum than the other feed treatments. Álvarez-Rodríguez et al. [43] indicated that lower concentrations of VFAs may be associated with lower TSP, as found in the current study in lambs fed 100% sprouted barley (T4), which is essential for rumen development [44].

In the current study, a strong positive correlation was observed between lactate and TVFA with papilla height, while a negative correlation was observed between format and propionate with papilla height and between format and total area papilla. When the VFA produced exceeds the absorption capacity of the rumen papillae, they accumulate in the rumen, resulting in lower pH in the rumen and destruction of the rumen epithelium [45]. 

## 5. Conclusions

In summary, replacing traditional feed with sprouted barley improved digestibility, rumen histomorphometric and increased the concentration of some VFAs and rumen bacteria, but resulted in a decrease in DMI and OMI, which negatively affected live weight gain. However, it should be noted that sprouted barley should not be used as the sole feed for lambs due to its very low content DM. Further studies are advisable to achieve optimal performance. 

## Figures and Tables

**Figure 1 vetsci-10-00015-f001:**
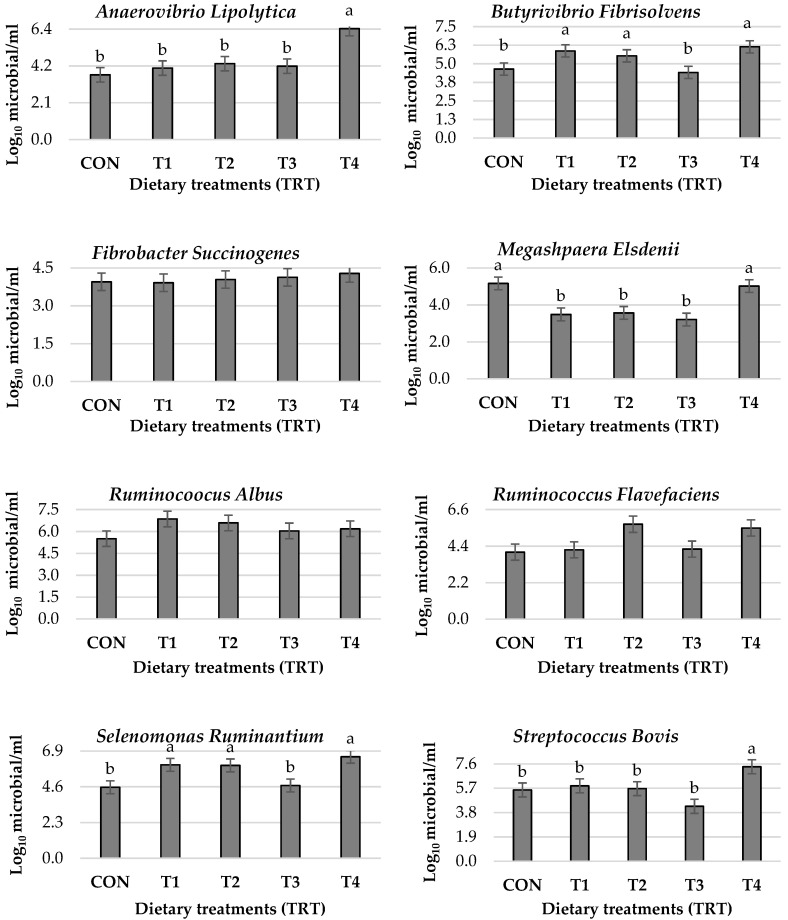
Effect of dietary treatments on ruminal bacterial quantification of growing lambs. Dietary treatments, CON = traditional diet 100% (barley 70%: alfalfa hay 30%); T1 = traditional diet 75%: sprouted barley 25%; T2 = traditional diet 50%: sprouted barley 50%; T3 = traditional diet 25%: sprouted barley 75% and T4 = sprouted barley 100%. ^a,b^ Means values within rows for each item with clarification of the significant difference in the form of superscripts (*p* < 0.05). *Anaerovibrio Lipolytica* (*p*-value: TRT = 0.002; L = 0.035; Q = 0.067). *Butyrivibrio Fibrisolvens* (*p*-value: TRT = 0.029; L = 0.080; Q = 0.908). *Fibrobacter Succinogenes* (*p*-value: TRT = 0.943; L = 0.718; Q = 0.782). *Megashpaera Elsdenii* (*p*-value: TRT = 0.001; L = 0.002; Q ≤ 0.0001). *Ruminocoocus Albus* (*p*-value: TRT = 0.452; L = 0.141; Q = 0.187). *Ruminococcus Flavefaciens* (*p*-value: TRT = 0.057; L = 0.128; Q = 0.707). *Selenomonas Ruminantium* (*p*-value: TRT = 0.010; L = 0.015; Q = 0.819). *Streptococcus Bovis* (*p*-value: TRT = 0.015; L = 0.701; Q = 0.082). Total bacteria (*p*-value: TRT = 0.5003; L = 0.720; Q = 0.333).

**Figure 2 vetsci-10-00015-f002:**
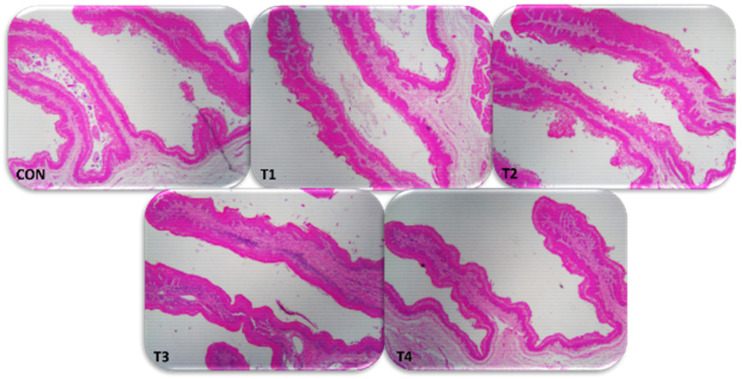
Photomicrographs to histomorphometric for rumen sections of growing lambs stained with hematoxylin and eosin (20×). CON = traditional diet 100% (Barley 70%: Alfalfa hay 30%); T1 = traditional diet 75%: sprouted barley 25%; T2 = traditional diet 50%: sprouted barley 50%; T3 = traditional diet 25%: sprouted barley 75% and T4 = sprouted barley 100%.

**Table 1 vetsci-10-00015-t001:** Ingredients and nutrient analysis as a % on a dry matter basis of dietary treatments ^1^.

Ingredients, %	Dietary Treatments (TRT)				
	CON	T1	T2	T3	T4
Barley grain	70.0	52.5	35.0	17.5	0.0
Alfalfa hay	30.0	22.5	15.0	7.5	0.0
Sprouted barley	0.0	25.0	50.0	75.0	100.0
Total	100	100	100	100	100
**Nutrient composition, %**					
Dry matter	95.5	76.6	57.8	38.9	20.1
Crude protein	15.0	14.7	14.4	14.2	13.9
Neutral detergent fiber	34.2	34.9	35.5	36.2	36.9
Acid detergent fiber	19.8	19.1	18.3	17.5	16.8
Lignin	5.8	4.9	4.0	3.0	2.1
Non-fibrous carbohydrates	43.9	43.8	43.7	43.6	43.5
Fat	1.7	1.9	2.2	2.4	2.7
Ash	5.2	4.6	4.1	3.5	3.0
Calcium	0.6	0.5	0.4	0.3	0.2
Phosphorus	0.2	0.2	0.3	0.3	0.4
Magnesium	0.2	0.2	0.2	0.1	0.1
Potassium	1.5	1.2	1.0	0.7	0.5
Sulfur	0.2	0.2	0.2	0.2	0.2
Sodium	0.2	0.2	0.2	0.2	0.2
Zinc, ppm	33.0	42.2	51.5	60.7	70.0
Copper, ppm	5.0	5.2	5.5	5.7	6.0
Net energy, Mcal/kg	1.85	1.87	1.88	1.89	1.91

^1^ Nutrient analysis of dietary treatments was performed in duplicate.

**Table 2 vetsci-10-00015-t002:** Primer sequences of ruminal bacterial genes for RT- qPCR analysis.

Target Gene	Forward (F) and Reverse (R) Primer (5′ → 3′)	Product Length	GenBank Number ^1^
*Anaerovibrio Lipolytica*	F: CACCAAGGCGACGATCAGTA R: CTGCCTCCCGTAGGAGTTTG	86	AB034191.1
*Butyrivibrio Fibrisolvens*	F: AGTAACGCGTGGGTAACCTG R: AAATCATGCGATTCCGTGCG	89	U41167.1
*Fibrobacter Succinogenes*	F: GTGCAAGCGTTGTTCGGAAT R: TCTACGCATTCCACCGCTAC	163	AB275514.1
*Megashpaera Elsdenii*	F: CGGCTACATTTCCCCGTACA R: GCGGTCCGTAATGAGGATGT	85	AY608424.1
*Ruminocoocus Albus*	F: AAAGAAGAAAGGCGGAGCGA R: CAGGCTTCGGCTCGATATGT	143	CP002407.1
*Ruminococcus Flavefaciens*	F: CGTTACCGCCCTTTCCTGAT R: AGGACGGCAAGCAATGAGAA	74	NC_001758.1
*Selenomonas Ruminantium*	F: GGTCTGAGAGGATGAACGGC R: CGAGCCGAAACCCTTCTTCA	137	M62703.1
*Streptococcus Bovis*	F: GCCGGTCTGAGAGGATGAAC R: AGACTTTCGTCCATTGCGGA	99	NR_113306.1
Total bacteria	F: GTGSTGCAYGGYTGTCGTCA R: ACGTCRTCCMCACCTTCCTC	130	NR_2828033

^1^ Design and test primers for this sequence using Primer-BLAST according to Szulc et al. [22].

**Table 3 vetsci-10-00015-t003:** Effect of dietary treatments on growth performance and apparent total tract digestibility in growing lambs from 1 to 75 days during the study period.

Parameters ^1^	Dietary Treatments (TRT) ^2^						*p*-Value ^4^		
	CON	T1	T2	T3	T4	SEM ^3^	TRT	L	Q
LWG (kg)	13.18 ^a^	10.60 ^a^	10.50 ^a^	10.35 ^a^	2.90 ^b^	1.46	0.009	0.010	0.18
DMI (kg/day)	1.46 ^a^	1.31 ^a^	0.95 ^b^	0.77 ^b^	0.23 ^c^	0.09	<0.0001	<0.0001	0.11
OMI (kg/day)	1.38 ^a^	1.25 ^a^	0.91 ^b^	0.74 ^b^	0.22 ^c^	0.09	<0.0001	<0.0001	0.10
Apparent total tract digestibility (%)									
DM	81.8 ^b^	81.6 ^b^	87.3 ^a^	88.1 ^a^	90.9 ^a^	1.69	0.004	0.010	0.88
OM	82.7 ^b^	82.5 ^b^	88.3 ^a^	88.9 ^a^	91.5 ^a^	1.63	0.004	0.010	0.95

^a–c^ Means values within rows for each item with clarification of the significant difference in the form of superscripts (*p* < 0.05). ^1^ LWG = Live weight gain; DMI = dry matter intake; OMI = organic matter intake. ^2^ Dietary treatments, CON = traditional diet 100% (barley 70%: alfalfa hay 30%); T1 = traditional diet 75%: sprouted barley 25%; T2 = traditional diet 50%: sprouted barley 50%; T3 = traditional diet 25%: sprouted barley 75% and T4 = sprouted barley 100%. ^3^ SEM = Standard error of means for diets effect. ^4^ TRT = treatments effect; L = linear response; Q = quadratic response.

**Table 4 vetsci-10-00015-t004:** Effect of dietary treatments on rumen pH and rumen color measurements of growing lambs.

	Dietary Treatments (TRT) ^2^						*p*-Value ^4^		
Parameters ^1^	CON	T1	T2	T3	T4	SEM ^3^	TRT	L	Q
Day 45									
pH0	7.0	7.3	7.1	7.3	6.9	0.22	0.61	0.72	0.22
pH3	6.4	6.4	6.8	6.5	6.2	0.15	0.11	0.48	0.02
pH different	0.7	0.8	0.4	0.7	0.8	0.14	0.48	0.78	0.56
Day 75									
pH0	6.5	6.2	6.3	6.4	6.4	0.18	0.85	0.40	0.34
pH3	6.2	6.1	6.1	5.9	6.3	0.18	0.57	0.81	0.24
pH different	0.4	0.2	0.4	0.5	0.2	0.12	0.40	0.56	0.57
Rumen Color									
L*	57.3	61.2	61.1	59.5	52.4	2.5	0.12	0.68	0.01
a*	1.5	2.0	0.8	1.5	0.4	0.4	0.15	0.56	0.42
b*	28.3 ^a^	29.2 ^a^	27.0 ^a^	23.2 ^ab^	18.1 ^b^	2.0	0.03	0.09	0.07

^a,b^ Means values within rows for each item with clarification of the significant difference in the form of superscripts (*p* < 0.05). ^1^ L = lightness; a = redness; b = yellowness; pH different = difference between pH0 and pH3 value. ^2^ Dietary treatments, CON = traditional diet 100% (barley 70%: alfalfa hay 30%); T1 = traditional diet 75%: sprouted barley 25%; T2 = traditional diet 50%: sprouted barley 50%; T3 = traditional diet 25%: sprouted barley 75% and T4 = sprouted barley 100%. ^3^ SEM = Standard error of means for diets effect. ^4^ TRT = treatments effect; L = linear response; Q = quadratic response.

**Table 5 vetsci-10-00015-t005:** Effect of dietary treatments on rumen volatile fatty acid concentrations of growing lambs.

	Dietary Treatments (TRT) ^2^						*p*-Value ^4^		
Parameters	CON	T1	T2	T3	T4	SEM ^3^	TRT	L	Q
Lactic acid (%)	24.2 ^a^	11.7 ^c^	10.1 ^c^	16.1 ^b^	3.4 ^d^	1.3	<0.0001	<0.0001	0.450
Formic acid (C1; %)	0.6 ^c^	1.9 ^b^	2.5 ^b^	1.7 ^b^	6.9 ^a^	0.3	<0.0001	<0.0001	0.006
Acetic acid (C2; %)	29.5 ^c^	37.2 ^b^	47.3 ^a^	37.2 ^b^	44.8 ^a^	1.7	<0.0001	<0.0001	0.038
Propionic acid (C3; %)	35.8 ^a^	32.3 ^a^	23.0 ^b^	24.6 ^b^	34.1 ^a^	1.7	<0.0001	0.001	<0.0001
Butyric Acid (C4; %)	9.9 ^c^	16.7 ^b^	16.9 ^b^	20.2 ^a^	10.7 ^c^	0.9	<0.0001	<0.0001	<0.0001
C2/C3 ratio	0.8 ^c^	1.1 ^bc^	2.1 ^a^	1.5 ^b^	1.4 ^b^	0.1	<0.0001	<0.0001	0.005
TVFA (mmol/mL) ^1^	0.184 ^a^	0.141 ^b^	0.114 ^c^	0.141 ^b^	0.077 ^d^	0.01	<0.0001	<0.0001	0.763

^a–d^ Means values within rows for each item with clarification of the significant difference in the form of superscripts (*p* < 0.05). ^1^ TVFA = total volatile fatty acid. ^2^ Dietary treatments, CON = traditional diet 100% (barley 70%: alfalfa hay 30%); T1 = traditional diet 75%: sprouted barley 25%; T2 = traditional diet 50%: sprouted barley 50%; T3 = traditional diet 25%: sprouted barley 75% and T4 = sprouted barley 100%. ^3^ SEM = Standard error of means for diets effect. ^4^ TRT = treatments effect; L = linear response; Q = quadratic response.

**Table 6 vetsci-10-00015-t006:** Effect of dietary treatments on the rumen histomorphometric of growing lambs.

	Dietary Treatments (TRT) ^2^						*p*-Value ^4^		
Parameters ^1^	CON	T1	T2	T3	T4	SEM ^3^	TRT	L	Q
Papilla height (mm)	1.64 ^c^	1.73 ^bc^	1.90 ^ab^	1.94 ^a^	1.34 ^d^	0.06	<0.0001	0.219	<0.0001
Papilla width (mm)	0.28 ^c^	0.30 ^bc^	0.33 ^a^	0.32 ^ab^	0.28 ^c^	0.01	<0.0001	0.002	<0.0001
SA (mm)^2^	1.45 ^c^	1.66 ^b^	2.01 ^a^	1.98 ^a^	1.18 ^d^	0.07	<0.0001	0.001	<0.0001
Density (n/cm^2^)	60.5 ^ab^	66.8 ^a^	61.6 ^ab^	55.0 ^b^	65.3 ^a^	2.51	0.014	0.574	0.510
TSP (mm^2^/cm^2^)	56.4 ^b^	70.9 ^a^	78.9 ^a^	69.6 ^a^	49.7 ^b^	3.91	<0.0001	0.016	<0.0001
SC (mm)	0.03 ^b^	0.03 ^b^	0.03 ^b^	0.03 ^b^	0.04 ^a^	0.001	<0.0001	0.216	0.196
WE (mm)	0.08 ^c^	0.09 ^b^	0.11 ^a^	0.09 ^b^	0.10 ^b^	0.002	<0.0001	<0.0001	<0.0001
LP (mm)	0.10 ^c^	0.12 ^bc^	0.13 ^b^	0.17 ^a^	0.11 ^c^	0.005	<0.0001	<0.0001	<0.0001
SM (mm)	0.11 ^e^	0.14 ^d^	0.20 ^b^	0.23 ^a^	0.18 ^c^	0.006	<0.0001	<0.0001	<0.0001

^a–e^ Means values within rows for each item with clarification of the significant difference in the form of superscripts (*p* < 0.05). ^1^ SA = Papilla surface area; TSP = The total surface of papillae; SC = stratum corneum; WE = the width of epithelium; LP = lamina propria; SM = Submucosa. ^2^ Dietary treatments, CON = traditional diet 100% (barley 70%: alfalfa hay 30%); T1 = traditional diet 75%: sprouted barley 25%; T2 = traditional diet 50%: sprouted barley 50%; T3 = traditional diet 25%: sprouted barley 75% and T4 = sprouted barley 100%. ^3^ SEM = Standard error of means for diets effect. ^4^ TRT = treatments effect; L = linear response; Q = quadratic response.

**Table 7 vetsci-10-00015-t007:** Pearson Correlation between volatile fatty acid and pH values or rumen histometeric.

	Papilla Height		TSP		pH Values	
Ruminal Bacteria Metabolites	Correlation Coefficient					
	rxy	*p* Value	rxy	*p* Value	rxy	*p* Value
Lactic acid	0.563	0.003	0.088	0.675	−0.012	0.927
Formic acid	−0.680	0.002	−0.413	0.039	0.246	0.235
Acetic acid	−0.179	0.391	0.242	0.242	0.104	0.618
Propionic acid	−0.487	0.013	−0.359	0.077	0.040	0.846
Butyric Acid	0.388	0.055	0.176	0.399	−0.323	0.114
TVFA	0.589	0.001	0.343	0.092	−0.184	0.378

TSP = The total surface of papillae; TVFA = total volatile fatty acid.

## Data Availability

The data and analyses presented in this paper are freely available with corresponding author on reasonable request.

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
