# Peer review of "Effects of Feeding Different Levels of Sprouted Barley on Fermentation Characteristics, Bacterial Quantification, and Rumen Morphology of Growing Lambs"

_vetsci, 2022, doi:10.3390/vetsci10010015_

Round 1
Reviewer 1 Report
All work must undergo an extensive grammatical review, the text is confusing and difficult to read. This makes it difficult for readers to appreciate the message the manuscript is carrying.
In saying this, I have outlined a couple of comments in the attached pdf document that needs to be looked at to improve the readability of the manuscript.

Author Response
Thanks for your valuable comments and our responses for them are attached

Reviewer 2 Report
The study fits the aims and scope of the journal. I would recommend its potential consideration after a thorough and major revision. Overall, I cannot always see why authors determined certain variables, e.g. ruminal tissue color or why the presence of a lypolytic bacterium was quantified when diets for ruminants only have very few fat contents. Also, the statistics is not sufficient, I think – please see comments below. This reduces the merit of the study, in my opinion.
· Hypothesis needs to be clear and is yet weak. What is an “improved fermentation”? More acetate, less methane? The hypothesis has to be verifiable
· L25: morphology of what?
· L30: for all levels of sprouted barley?
· L31-33: Microbial species have to be written in italics
· L59: How can the feedstuff itself support feed digestion?
· Overall, the text transitions should be written more nicely.
· L74: How is it improving health?
· Table 1: Why is the nutrient concentration in T2 often lower/higher than in other diets? For instance NDF, NFC, Net energy
· L115: What is meant by 5%?
· Growth performance: Only weighing in the beginning and at the end is a bit weak. If authors would weigh more often, they can see potential week effects + which devices were used?
· The digestibility you measured it apparent total tract digestibility (ATTD), correct? This has to be worded like this!
· L140: Which company from Augsburg, Germany?
· State the name of the DNA extraction kit
· L153: 10,000 × g
· Table 2: Include the references for the primers + why is there two times M. elsdenii?
· Why did authors not quantify total bacteria?
· L169: 2 cm is a lenth, please state the area.
· L186: Name the factors that were included in the model. Why did authors not use a mixed model and accounted for a random effect of animal? Then authors could have also used repeated measurement statements, which is missing but essential in my opinion. Please re-analyze.
· Table 3: Why is T4 suddenly so bad? Authors mention the low DM content, but T3 is still fine and has a lower DM content, too. Was the feed quickly spoiled?
· Table 5: Please provide TVFA in mmol/ml.
· Figure 1: The scales should be uniform for all bacterial species + why did authors decide for those (and not others)?
· L352: I think it is rather because there is less DM to ferment in the rumen
· L360: Higher ruminal degradability means more VFA production. So is this true?
· L377: But NDF is similar for T4 and CON and more or less for all diets. So this cannot be the reason here
· L421: replacement of traditional feed with sprouted barley
Author Response

(The authors gave the same response as above.)

Round 2
Reviewer 1 Report
Most major queries from my first review has been resolved. Attached is the pdf of a few minor corrections to improve readability.

Author Response
Dear Reviewer 1,
On behalf of my colleagues, we would like to thank you for considering our manuscript
entitled "Effects of Feeding Different Levels of Sprouted Barley on Fermentation
Characteristics, Bacterial Quantification, and Rumen Morphology of Growing Lambs".
We have revised the manuscript based on the reviewers’ comments point-by point and
our responses to the reviewer are marked up using the “Track Changes” each specific
comment in round 1 whereas our using the “Track Changes” with Text Highlight
Yellow Color each specific comment in round 2
We hope the changes made are satisfactory to Your Excellency and to the respected
reviewers.

Reviewer 2 Report
The R1 version made some improvements.
Further aspects from my side:
· It has to be “apparent total tract digestibility”. Only apparent digestibility would mean that authors did not correct for endogenous losses, which is true – but they also measured total tract. This has to be corrected
· Change “general bacteria” to “total bacteria” in primer table
· I am skeptical about the theory of higher NDF causing changes in T4 compared to CON. The difference is very small and not statistically or biologically relevant, I believe
· The statistics is not strong. Of course authors used lambs from the same breed etc. but inter-individual differences between animals are always present. So this should be considered during analysis. Plus, authors did repeated measurements for pH determination, I think
· I was surprised that authors suddenly had total bacteria qPCR data, but did not show it before
Author Response
Dear Reviewer 2,
On behalf of my colleagues, we would like to thank you for considering our manuscript
entitled "Effects of Feeding Different Levels of Sprouted Barley on Fermentation
Characteristics, Bacterial Quantification, and Rumen Morphology of Growing Lambs".
We have revised the manuscript based on the reviewers’ comments point-by point and
our responses to the reviewer are marked up using the “Track Changes” each specific
comment in round 1 whereas our using the “Track Changes” with Text Highlight
Yellow Color each specific comment in round 2
We hope the changes made are satisfactory to Your Excellency and to the respected
reviewers.

Round 3
Reviewer 2 Report
I thank the authors for again revising their manuscript.